# Interatrial Block, Bayés Syndrome, Left Atrial Enlargement, and Atrial Failure

**DOI:** 10.3390/jcm12237331

**Published:** 2023-11-26

**Authors:** Roberto Bejarano-Arosemena, Manuel Martínez-Sellés

**Affiliations:** 1Cardiology Department, Hospital General Universitario Gregorio Marañón, 28007 Madrid, Spain; robertoandres.bejarano@salud.madrid.org; 2Centro de Investigación Biomédica en Red de Enfermedades Cardiovasculares (CIBERCV), Instituto de Salud Carlos III, 28029 Madrid, Spain; 3School of Health and Biomedical Sciences, Universidad Europea, 28670 Madrid, Spain; 4School of Medicine, Universidad Complutense, 28040 Madrid, Spain

**Keywords:** interatrial block, Bayés syndrome, left atrial enlargement, atrial failure, stroke, atrial fibrillation, anticoagulation

## Abstract

Interatrial block (IAB) is defined by the presence of a P-wave ≥120 ms. Advanced IAB is diagnosed when there is also a biphasic morphology in inferior leads. The cause of IAB is complete block of Bachmann’s bundle, resulting in retrograde depolarization of the left atrium from areas near the atrioventricular junction. The anatomic substrate of advanced IAB is fibrotic atrial cardiomyopathy. Dyssynchrony induced by advanced IAB is frequently a trigger and maintenance mechanism of atrial fibrillation (AF) and other atrial arrhythmias. Bayés syndrome is characterized by the association of advanced IAB with atrial arrhythmias. This syndrome is associated with an increased risk of stroke, dementia, and mortality. Advanced IAB frequently produces an alteration of the atrial architecture. This atrial remodeling may promote blood stasis and hypercoagulability, triggering the thrombogenic cascade, even in patients without AF. In addition, atrial remodeling may ultimately lead to mechanical dyssynchrony and enlargement. Atrial enlargement is usually the result of prolonged elevation of atrial pressure due to various underlying conditions such as IAB, diastolic dysfunction, left ventricular hypertrophy, valvular heart disease, hypertension, and athlete’s heart. Left atrial enlargement (LAE) may be considered present if left atrial volume indexed to body surface is > 34 mL/m^2^; however, different cut-offs have been used. Finally, atrial failure is a global clinical entity that includes any atrial dysfunction that results in impaired cardiac performance, symptoms, and decreased quality of life or life expectancy.

## 1. Introduction

In recent years, the atria have been studied in greater depth, given their important role in global cardiac function and their association with different cardiovascular conditions. The most important arrhythmia associated with atrial disease is atrial fibrillation (AF), which increases the risk of thromboembolic events and stroke, the latter being the cause of more than 7 million deaths worldwide annually. Advances in electrocardiography, electrophysiology, and cardiac imaging, have made possible to obtain in-depth information on these cavities. This could help to detect the atriopathy at an early stage and potentially prevent its deleterious clinical outcomes. New terms have been defined in relation to atrial disease which, while sharing common features, are clearly distinguishable, with different diagnostic and clinical characteristics. Due to the relevance of these entities, we believe that every physician should be familiar with expressions such as interatrial block (IAB) (both partial and advanced), Bayés syndrome, left atrial enlargement, and atrial failure.

## 2. Atrial Anatomy and Physiology

The atria are complex structures that play a fundamental role in cardiac function. The left atrium includes a posterior region where the blood flows from the four pulmonary veins and an anterior region that connects with the left ventricle through the mitral valve. Between these walls we find the left atrium appendage, a trabeculated structure of great anatomical variability, which has an endocrine function and, due to its morphology, is susceptible to alterations in blood flow that can lead to thrombus formation. In AF-related thromboembolism, left atrial appendage is the most common site of thrombus origin, representing 90% of the cases [1]. This provides the rationale for left atrial appendage closure (percutaneous or surgical), particularly in patients who have contraindications to anticoagulant therapy, to reduce stroke risk.

Electrically, the atria are connected to each other by fast conduction bundles formed by specialized myocardial tissue, which can be intra- or interatrial. The sinoatrial node, the natural pacemaker of the heart, is connected to the atrioventricular node by three bundles—Bachmann’s bundle, the Wenchebach bundle, and the Thorel bundle—which are located in the right atrium anteriorly, medially, and inferiorly, respectively [2]. Bachmann’s bundle also has a broad muscular extension that lies in the anterior interatrial septum and reaches the left atrial appendage, connecting both atria to maintain interatrial synchrony [2]. In addition to this main interatrial conduction pathway, there are other secondary pathways that are located near the right pulmonary veins (posterior pathways) and near the coronary sinus and the fossa ovalis (inferior pathways) [3]. The aforementioned circuits allow for normal electrical conduction between the two atria, resulting in normal P wave morphology. Therefore, impairment of any one of these pathways will result in alterations in interatrial conduction and produce changes in the duration or morphology of the P wave.

The atrium contribute to ventricular function, especially during diastole. During ventricular systole and isovolumetric relaxation, the left atrium is filled by the incoming flow from the pulmonary veins (reservoir function). When the mitral valve opens, there is a rapid flow of blood into the left ventricle, approximately half of the cardiac output. While the mitral valve remains open, there is a passive flow of blood from pulmonary veins to left ventricle (conduit function), that represents approximately 20% of cardiac output. Finally, left atrial contraction allows the left ventricular filling to be completed at the end of diastole, contributing approximately 30% of cardiac output [4]. Reservoir and conduit function depend on atrial distensibility, ventricular relaxation, and mitral transvalvular gradient, so each of these factors, in addition to atrial contractile function, affects left atrial function [4].

## 3. Interatrial Block

Interatrial conduction is mainly mediated by Bachmann’s bundle, a large muscular band located in the anterior region of the interatrial septum. Anatomical and/or functional alterations of this bundle can cause conduction disturbances between the two atria leading to IAB, producing P wave changes [5] (Figure 1). IAB is clinically relevant as it is associated with the development of AF, stroke, cognitive impairment and dementia, especially in the case of advanced IAB [6,7].

### 3.1. History

Bachmann described the first case of IAB in 1941 [8]. In the late 1970s, Bayés de Luna described the types of intra- and interatrial blocks and, shortly thereafter, the prevalence of advanced IAB [8]. In 1988, the association of IAB with supraventricular arrhythmias, in particular with AF, was demonstrated [9]. In 2007, the association between IAB and the presence of atrial thrombus in patients with embolic stroke of unknown source was described [10]. The first consensus on this condition was published in 2012 [11] and in 2014, the term “Bayés syndrome” was coined to describe the association of advanced IAB with atrial tachyarrhythmias [12].

### 3.2. Definition

Partial (or first degree) IAB is due to a delay in interatrial conduction. The ECG shows a P wave ≥ 120 ms, positive morphology in the inferior leads. For correct measurement, it is important to consider the beginning of the P wave in the first lead in which it starts and the end in the last lead in which it terminates. Advanced (or third-degree) IAB occurs when conduction through Bachmann’s bundle is completely blocked (Figure 2), so that electrical conduction must occur through the minor bundles located below, in the fossa ovalis or coronary sinus, so that depolarization of the left atrium is caudocranial. Therefore, in addition to the aforementioned prolongation of the P wave, the ECG shows a biphasic P wave in inferior leads [13] (Table 1).

Atypical subtypes of advanced IAB have been recently described that also appear to be associated with the development of atrial arrhythmias [14,15] (Table 2). Finally, the intermittent form of this entity is known as second-degree IAB. It may progress from a normal P wave to fulfilling the criteria for IAB in the same rhythm strip, or it may alternate between partial IAB and advanced IAB [16].

### 3.3. Epidemiology

Partial IAB is common in the general population (20%), while advanced IAB is not (0.5%) [17,18]. The prevalence of IAB increases significantly with age, reaching up to 60% in patients over 65 years of age. In addition to age, other risk factors for its development include arterial hypertension, diabetes, obesity, myocardial ischemia, heart disease, sleep apnea, interventions such as catheter ablation and cardiac surgery [6,19,20,21,22,23].

### 3.4. Physiology and Imaging Testing

Asynchronous and delayed atrial contraction in patients with IAB increases atrial pressure and may facilitate the development of atrial fibrosis. Fibrosis plays a key role in IAB and can promote its development and/or be a consequence of IAB. The presence of diffuse fibrosis in the left atrium is common in patients with IAB [4,24], and the increase in extracellular matrix slows electrical conduction through the atria [25]. The best way to determine the presence and extent of atrial fibrosis is by cardiac magnetic resonance imaging with late gadolinium enhancement, although the detection of atrial fibrosis is still not well standardized. The concentration of this contrast agent in the myocardial tissue depends on the volume of the extracellular space. If the extracellular volume is increased, as in fibrosis, gadolinium will still be visible on late sequences. It has been shown that the presence and degree of late gadolinium enhancement is associated with a higher prevalence of IAB and AF [24,26].

Due to the limited availability and high cost of magnetic resonance imaging, other ways of detecting atrial fibrosis have been studied. With transthoracic echocardiography and speckle tracking, atrial deformation (global longitudinal strain) can be assessed which is usually diminished due to altered atrial distensibility [27]. In fact, a relationship between decreased atrial strain and the prevalence of IAB and AF has been observed [20,28]. Therefore, echocardiography may be an indirect method to detect atrial fibrosis, with a predictive power for the development of IAB and AF similar to that of magnetic resonance imaging [24]. Finally, functional echocardiography changes have been observed in IAB, as electromechanical latency (from the P-wave onset and the A’ echocardiography wave, which represents atrial contraction). A lower degree of atrial contribution to ventricular filling has also been observed, almost similar to that seen in AF [4].

## 4. Bayés Syndrome

The association between advanced IAB and atrial tachyarrhythmias, mainly AF, is called Bayés syndrome [12]. Advanced IAB and AF have common pathophysiologic basis closely related to atrial fibrosis [24,28]. Even in patients with paroxysmal AF, stroke is not temporally related to AF episodes, so the cause of cerebral infarction in these patients is the atrial damage that creates a prothrombotic environment rather than the arrhythmia itself, which we should see as a marker of atriopathy rather than a direct cause of stroke [29,30]. In addition, both IAB and AF produce flow alteration, particularly in left atrial appendage, which favors thrombus formation [28]. In fact, the two entities favor each other: AF causes greater atrial remodeling and fibrosis, which favors the development of IAB, and IAB causes electrical reentry in the Bachmann region and the generation of atrial extrasystoles, favoring the development of AF [31].

### 4.1. IAB and Prognosis

#### 4.1.1. AF and Stroke

IAB, particularly advanced IAB, is a strong predictor of AF. This association has been demonstrated in several clinical scenarios including heart failure, ischemic heart disease, Takotsubo syndrome, transcatheter aortic valve implantation, cardiac amyloidosis, sleep apnea, cavotricuspid isthmus and pulmonary vein ablation, and following electrical/pharmacological cardioversion [32,33,34,35,36,37,38]. However, the predictive power of IAB appears to be lower in other situations, such as severe chronic kidney disease [39]. IAB is also associated with stroke risk [40] and this association is independent of the presence of AF [41] as could be expected due to the aforementioned fibrotic atriopathy that causes electromechanical dissociation. The BAYES (interatrial block and yearly events) registry was an international prospective study of 556 patients ≥70 years with structural heart disease and sinus rhythm, with a median follow-up of 694 days. The authors found that advanced IAB was independently associated with AF (hazard ratio 2.9, 95% confidence interval 1.7–5.1; *p* < 0.001) and stroke (hazard ratio 3.8, 95% confidence interval 1.4–10.7; *p* = 0.010) [41]. Moreover, advanced IAB is often associated with atrial dilatation [42], favoring blood stasis, hypercoagulability, and thrombus formation in the atrial appendage (Figure 3). P-wave indices indicative of atrial fibrosis, such as IAB, prolonged P-wave duration, abnormal P-wave axis, and abnormal P-wave terminal force in lead V1, predict the development of AF and consequent stroke risk. Therefore, it has been proposed to incorporate these markers into classic stroke risk scores such as CHA_2_DS_2_-VASc to improve their ability to predict AF-related ischemic stroke [43,44]. A study of 2929 patients with incident AF found that of all P-wave indices, abnormal P-wave axis was the most important predictor of ischemic stroke (hazard ratio, 1.84; 95% confidence interval, 1.33–2.55) independent of CHA_2_DS_2_-VASc variables, followed by advanced IAB, so the authors concluded that adding 2 points for abnormal P-wave axis to the CHA_2_DS_2_-VASc score (P_2_-CHA_2_DS_2_-VASc) improved its stroke prediction [43]. Thus, the authors suggest that P-wave indices could be considered to guide anticoagulation in patients with AF. The role of IAB has been studied in embolic stroke of undetermined source [45] as it significantly increases the prevalence of atrial appendage thrombus (15% vs. 0% in patients with normal P wave) [10]. In fact, patients with embolic stroke of undetermined origin have as much atrial fibrosis as those with AF [26]. This also suggests that fibrotic atriopathy is a cause of stroke independent of AF, considering that this arrhythmia is detected during follow-up in only 30% patients with embolic stroke of undetermined source [46,47]. Attempts are being made to use electrocardiographic, echocardiographic, or biomarker parameters suggestive of fibrotic atriopathy to identify which of these patients might benefit from anticoagulation. However, to date, these efforts have been unsuccessful, probably due to inadequate definition of atriopathy in clinical trials [48,49,50,51].

#### 4.1.2. Dementia and Mortality

IAB is associated with cognitive impairment, perhaps due to micro-embolism vascular causes, although other mechanisms might be involved [52]. In the registry Scientific Characterization of the Centennial Heart—Caracterización Científica del Corazón del Centenario (4C), the prevalence of dementia was associated with IAB and its degree, being 70% in partial IAB, 80% in advanced IAB, and 90% in AF [18]. The prospective registry BAYES showed an independent association between IAB and the development of dementia at follow up of 22 months (hazard ratio 2.01, 95% confidence interval 1.25–3.22) [7]. This association was more pronounced in patients with advanced IAB (hazard ratio 2.04, 95% confidence interval 1.19–3.51, *p* = 0.01, for advanced IAB). In addition to IAB, other atriopathy-related P-wave indices have been shown to be associated with incident dementia [53]. IAB is also associated with increased mortality. P-wave duration has been associated with sudden death, cardiovascular death, and all-cause death, independent of other cardiovascular risk factors and the presence of AF [40]. In addition, the BAYES registry demonstrated that P-wave duration was independently associated with all-cause mortality (hazard ratio 1.04, 95% confidence interval 1.00–1.08; *p* = 0.021) [41]. It has been observed that patients with an IAB are at an increased risk for ventricular arrhythmias, both in general population and in patients with dilated cardiomyopathy [54,55]. Although the mechanism of the association of IAB with sudden death is not well understood, the finding of diffuse ventricular fibrosis on magnetic resonance imaging in patients with advanced IAB could be one of the explanations [56].

### 4.2. IAB Management

The basis of treatment is to target and address the risk factors associated with the fibrotic atriopathy. For instance, the use of continuous positive airway pressure in patients with obstructive sleep apnea shortens the duration of the P wave and reduces the recurrence of AF [57]. Anti-remodeling drugs used in heart failure that reduce the development of ventricular fibrosis, which may also have an impact on atrial fibrosis, although no clear evidence about this is available yet [58]. Treating interventricular dyssynchrony with cardiac resynchronization therapy decreases P-wave duration and reduces the incidence of IAB [59], although there is insufficient evidence to recommend cardiac resynchronization therapy on this basis alone. Antiarrhythmics could reduce the risk of developing AF in patients with IAB. In a small study of 32 patients with advanced IAB and structural heart disease, 94% of patients without antiarrhythmic treatment had at least one episode of supraventricular tachyarrhythmia during 32 months of follow-up, compared with 25% of patients who developed AF while on antiarrhythmic drugs [60]. However, given the low level of evidence and the side effects of antiarrhythmic drugs, their use for this purpose cannot be recommended.

Probably the most important aspect of IAB management is the identification of patients with IAB who are at highest risk for stroke [61,62,63]. Patients with advanced IAB, very prolonged P-wave duration, structural heart disease, frequent atrial extrasystoles, and elevated CHA_2_DS_2_-VASc score are at highest risk of stroke [64] (Table 3). However, prospective studies confirming the association between IAB and the development of stroke, such as the BAYES registry [65] and the Rigorous Atrial Analysis in Advanced Interatrial Block—Análisis Riguroso Auricular en Bloqueo Interauricular Avanzado (ARABIA) study [66], are not sufficient to make firm anticoagulation recommendations. A randomized clinical trial of anticoagulation versus placebo would be needed to be able to show anticoagulation benefit in this setting [67]. In the meantime, we can only recommend a close and intensive follow-up of patients with advanced IAB to detect AF promptly and initiate anticoagulation in time. The goal is to reduce the occurrence of stroke and its associated morbidity, mortality, and hospital costs [67].

## 5. Left Atrial Enlargement

Atrial enlargement or dilation is usually the result of prolonged elevation of atrial pressure due to various underlying conditions such as diastolic dysfunction, left ventricular hypertrophy, valvular heart disease, hypertension, and athlete’s heart. Patients with atrial dysfunction due to atriopathy, such as those with IAB, may also experience atrial enlargement.

LAE has been associated with adverse cardiovascular events, including AF, stroke, and mortality. This association appears to be more robust in patients with diabetes mellitus, dilated cardiomyopathy, valvular heart disease, and after acute myocardial infarction [68,69,70,71].

### 5.1. Diagnosis

The diagnosis is made with imaging tests, most frequently transthoracic echocardiography, although computed tomography and magnetic resonance imaging are also useful [72]. The most used echocardiographic method was the antero-posterior diameter in a parasternal long-axis view. However, this does not accurately represent atrial size as it assumes that when the atrium grows, it does so equally in all dimensions [73]. Therefore, its isolated use is not recommended. Currently, the most recommended method is left atrium volume measurement, due to its better prognostic value [70,74]. The method of summation of disks in two orthogonal planes (apical two-chamber and four-chamber views) is commonly used, similar to ventricular volumes estimation (Figure 4). Sex difference in atrial size is attenuated when adjusted for body surface area; therefore, it is currently recommended that only indexed measurements should be reported [75]. Although different cut-offs have been used for years, several studies have attempted to establish normal values for left atrial volume, with an indexed volume of 34 mL/m^2^ being accepted as the maximum normal value [74,75,76].

### 5.2. Left Atrial Enlargement and Interatrial Block

Although P wave duration and IAB are associated with LAE, the electrocardiogram cannot be used as the sole diagnostic test to assess LAE, due to its low sensitivity and specificity [77,78]. It has been suggested that other electrocardiographic markers, such as the large negative component in V1, may be more specific for LAE [11]. For this reason, a consensus published in 2009 recommended the term “left atrial anomaly” to include both LAE and IAB [79]. It is clear, however, that both entities can exist separately, although they can coexist. A 2012 consensus emphasized that these are two different entities [11] and several reasons justify this statement. First, as we have seen, IAB pattern can be transient, even within the same rhythm strip. Second, approximately half of partial IAB patients have normal-sized atria, and even if the association of advanced IAB with LAE is much higher, not all patients with advanced IAB have LAE [80,81]. In fact, the increase in P-wave duration in LAE is mainly due to underlying IAB than by the greater distance the electrical impulse must travel due to atrial dilatation. Third, it has been observed that LAE is common in athletes without being associated with P-wave changes or IAB [78]. Finally, it has been observed that IAB can be reproduced experimentally by ablation of Bachmann’s bundle, demonstrating that this is an exclusively electrical conduction phenomenon that might not be associated with LAE [82].

## 6. Atrial Failure

### 6.1. Definition

Although atriopathy can be a result of a variety of heart conditions, it can also be the cause of many of them [83]. In 2020, the term atrial failure was proposed as a clinical entity defined as “any atrial dysfunction (anatomical, mechanical, electrical, and/or rheological, including blood homeostasis) causing impaired heart performance and symptoms, and worsening quality of life or life expectancy, in the absence of significant valvular or ventricular abnormalities” [84]. Causes of atrial failure include arrhythmias and conduction disorders (such as IAB), fibrotic atrial cardiomyopathy, atrial remodeling in response to atrial volume/pressure overload or secondary to chronic atrial arrhythmias, such as AF [85].

### 6.2. Clinical Manifestations

Atrial failure facilitates the development of heart failure, pulmonary hypertension, and stroke independent of the presence of AF [86,87]. It also predisposes to the development of AF which in turn causes fibrosis of both the atrium and ventricle [88], perpetuating atriopathy. In addition, by an independent mechanism that is not fully elucidated, strokes also induce left atrium structural changes, such as fibrosis, inflammation, and endothelial dysfunction, which further contribute to atriopathy [89]. Atrial failure may activate neurohormonal compensatory mechanisms that exacerbate atrial dysfunction, similar to what occurs in cardiomyopathies, suggesting a future therapeutic target [90] (Figure 5).

Although the primary substrate of heart failure with preserved ejection fraction appears to be left ventricular diastolic dysfunction, many cases of heart failure with preserved ejection fraction are caused by left atrial dysfunction [91]. In fact, atrial reservoir and conduit function correlate with peak oxygen consumption [92] and atrial dysfunction/remodeling may precede the development of heart failure [91,93]. Up to 45% of patients who develop heart failure have altered atrial strain [93]. In addition to its potential to cause symptoms of heart failure, atrial dysfunction is associated with episodes of decompensation. In the early stages of ventricular dysfunction, left atrium adapts to the increased preload without increasing pulmonary capillary pressure. As ventricular dysfunction progresses, the atrium fails to accommodate the excess pressure, leading to postcapillary pulmonary hypertension and atrial failure syndrome [94].

Other changes associated with atrial failure include ventricular dysfunction secondary to AF with sustained rapid ventricular response, known as tachycardiomyopathy [95]. Another relevant consequence of rapid-response AF is its association with myocardial ischemia and acute type 2 myocardial infarction, either due to alteration of coronary flow in diastole or by an embolic mechanism [96]. Finally, atriopathy has been shown to cause mitral or tricuspid regurgitation due to dilatation and altered annular dynamics, that may cause or exacerbate heart failure symptoms, promote the development of AF, and influence prognosis [97,98].

### 6.3. Conclusions

Atrial failure should be considered in patients with compatible symptoms in the presence of atriopathy not attributable to ventricular or valvular disorders or extracardiac conditions. Advances in cardiac imaging and electrophysiology will allow us to improve our understanding of left atrial pathophysiology and its impact on global cardiac function. The new definition of atrial failure is likely to increase the volume of research focused on the left atrium in the future, with the goal of identifying therapeutic targets to prevent the clinical consequences of this entity.

## 7. Conclusions and Future Directions

IAB is a conduction disturbance between atria seen as a prolonged P-wave duration (≥120 ms). Advanced IAB has a biphasic morphology in inferior leads and is clearly associated with the risk of AF (Bayés syndrome), stroke, cognitive impairment, and mortality. LAE is defined as left atrial index volume > 34 mL/m^2^ and is also associated with cardiovascular events. Finally, any primary or secondary atrial alteration can eventually lead to atrial failure, which facilitates the onset of heart failure and its decompensation.

In the coming years, further advances in electrophysiology and cardiac imaging research will continue to provide us with more information about the importance of the atria. New tools to identify atriopathy in a timely manner are needed. In addition, research focused on effective treatments, particularly in patients with high stroke risk, as those with advanced IAB, might find new options to improve their prognosis.

## Figures and Tables

**Figure 1 jcm-12-07331-f001:**
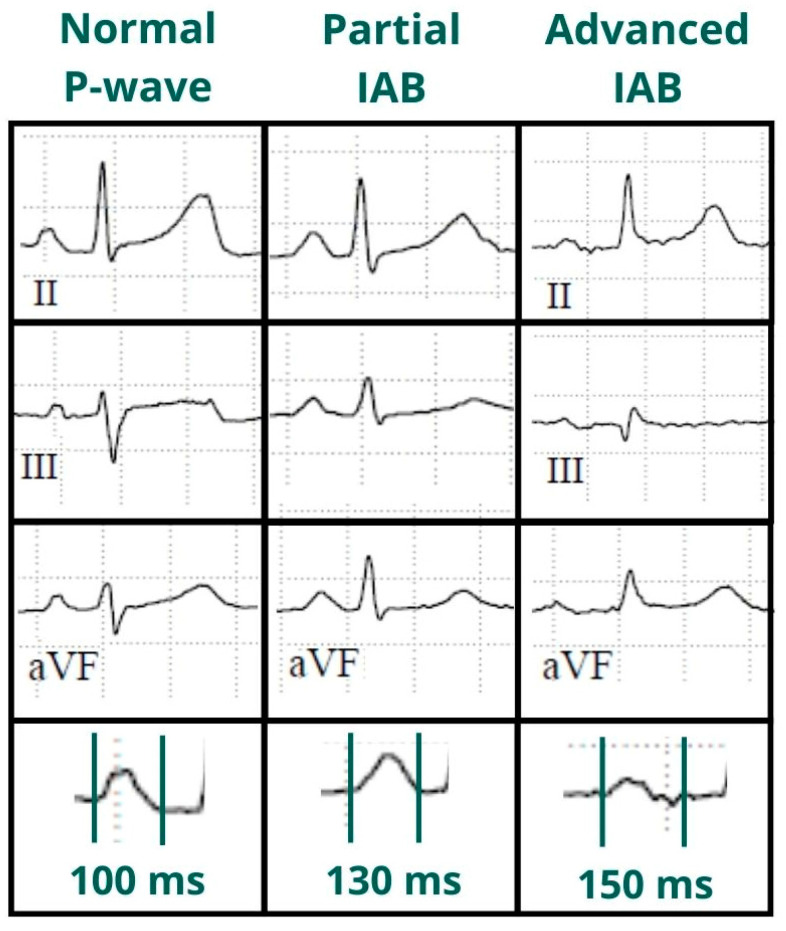
Normal P-wave, partial and advanced interatrial block (IAB). Normal P-wave has a duration < 120 ms. IAB has a P-wave duration ≥ 120 ms and in partial IAB is positive in inferior leads and has a biphasic morphology in advanced IAB.

**Figure 2 jcm-12-07331-f002:**
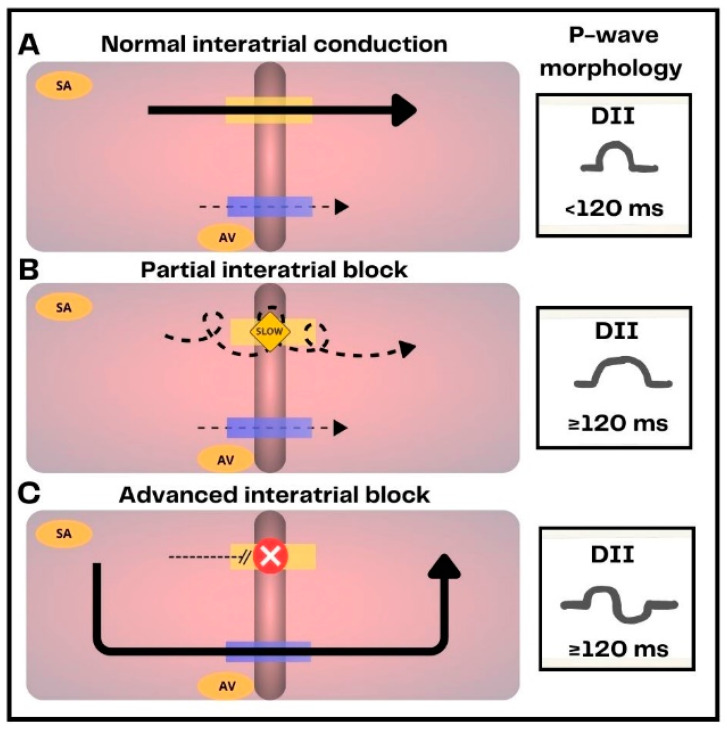
Representation of normal interatrial conduction, interatrial blocks (IAB) and their electrocardiographic expression. (**A**) In healthy subjects interatrial conduction is primarily mediated by Bachmann’s bundle (yellow bar). The electrical impulse is also conducted to a lesser extent through the coronary sinus tracts (blue bar). (**B**) In partial IAB, conduction through Bachmann’s bundle is delayed, resulting in prolonged P-wave duration (≥120 ms). (**C**) In advanced IAB, Bachmann’s bundle is completely blocked, the electrical impulse is conducted through inferior branches of the coronary sinus, producing left atrium caudo-cranial activation. This results in a prolonged P-wave duration with biphasic morphology.

**Figure 3 jcm-12-07331-f003:**
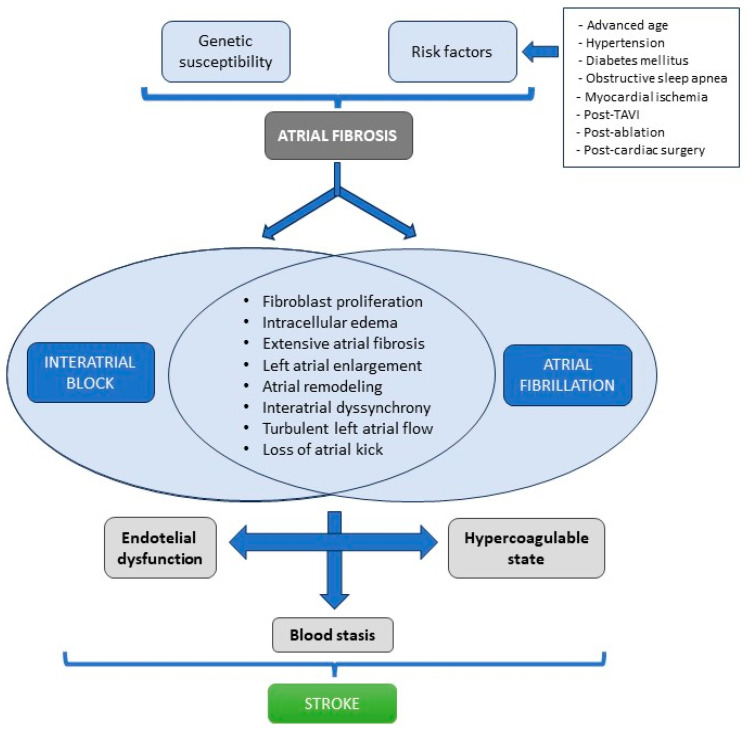
Interatrial block and atrial fibrillation induce changes that increase the risk of stroke.

**Figure 4 jcm-12-07331-f004:**
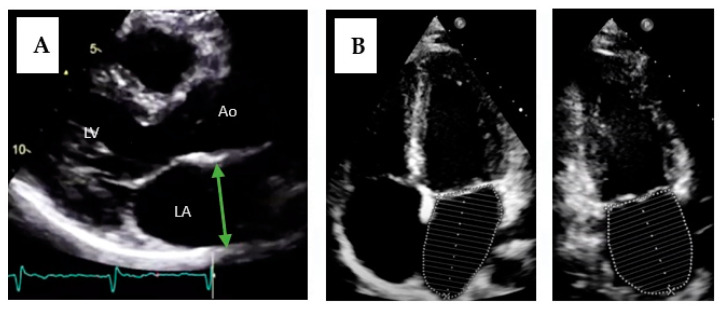
Measurement of left atrium size by transthoracic echocardiography. (**A**) Measurement of the antero-posterior diameter of the left atrium. LA = left atrium; LV = left ventricle; Ao = aorta. (**B**) Measurement of atrial volume using the disks method in two planes (four-chamber apical and two-chamber apical views).

**Figure 5 jcm-12-07331-f005:**
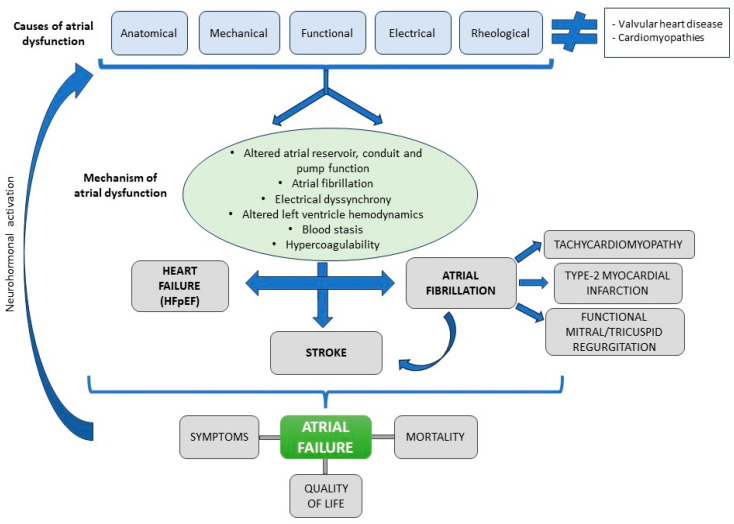
Atrial failure: causes, mechanisms, and clinical manifestations.

**Table 1 jcm-12-07331-t001:** Types of interatrial block.

Classification	Degree	Pathophysiology	ECG Pattern	Morphology
Partial	First	Delayed conduction through Bachmann’s bundle	P ≥ 120 ms + in inferior leads	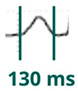
Intermittent	Second	Transient block of Bachmann’s bundle	Partial changes to advanced and viceversa	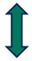
Advanced	Third	Bachmann’s bundle block. Left atrium activation in caudocranial direction.	P ≥ 120 ms ± in inferior leads	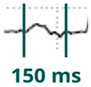

**Table 2 jcm-12-07331-t002:** Atypical types of advanced interatrial block.

Morphologic Criteria	ECG Features	Atypical Finding
Type 1	P ≥ 120 ms ± in III and aVF. Final P wave component is isoelectric in II	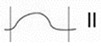
Type 2	P wave ≥ 120 ± in III and aVF. Final P wave component is triphasic in II	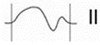
Type 3	P wave ≥ 120 ms ± in II and negative in III and aVF	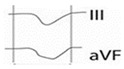
Durationcriteria	P wave < 120 ms ± in inferior leads	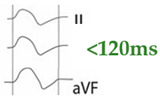

**Table 3 jcm-12-07331-t003:** Factors associated with a higher thromboembolic risk in patients with advanced interatrial block.

P wave duration ≥ 160 ms
Structural heart disease
>40 atrial extrasystole per hour or paroxysm of AT/AF in Holter recording
CHA_2_DS_2_-VASc ≥ 2

AT: atrial tachycardia; AF: atrial fibrillation. CHA_2_DS_2_-VASc: congestive heart failure, hypertension, age ≥ 75 years, diabetes mellitus, prior stroke/transient ischemic attack/thromboembolism, vascular disease, age 65–74 years, and sex category (female).

## Data Availability

No new data were created or analyzed in this study. Data sharing is not applicable to this article.

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
