# Peer review of "Interatrial Block, Bayés Syndrome, Left Atrial Enlargement, and Atrial Failure"

_jcm, 2023, doi:10.3390/jcm12237331_

Round 1

Reviewer 1 Report

Comments and Suggestions for Authors

This is an interesting and thorough review of a somewhat disregarded entity - interatrial block and Bayes syndrome. The authors should be proud of their work. 

Some minor corrections would improve the overall quality of this review:

 - The authors should refrain from adjectivaction / literary remarks. Some examples: line 70 "IAB is of great clinical importance (...), line 82 "undoubtedly demonstrated". 

 - For clarification, please rephrase line 115-116 "reaching up to 60% > 65 years". Also, line 139-140 should be grammatically reviewed. 

- In section 4.1, the impact of IAB and prognosis is addressed and the references add up. Some numbers could be shown - p.e., what does the literature says regarding the risk increase that IAB brings for AF or stroke? The same question can be made for dementia and mortality.

Also, in line 167 the authors mention that "It has been proposed to incorporate electrocardiographic markers of atrial fibrosis to the classic stroke risk scores such as CHA2DS2-VASC to improve their predictive power" -->but no conclusion regarding this topic is included. What was the impact that incorporating ECG markers in CHADS VASC had?

I would like to add that this review is so thorough that the authors should be  challenged to present a meta analysis quantifying the impact that IAB has as an independent risk factor for atrial arrythmias - in a future manuscript, at least.

- Please review line 208/209: "Antiarrhythmics reduce the risk of developing AF in patients with IAB, in a small study 25% develop AF in the treatment group vs. 90% in the control group) [94]." Also line 212, "Perhaps the most important aspect could be~the identification..." and line 221 "In the meantime we can only recommend is close and intensive"... This paragraph should be reviewed.

- Line 242 regarding methods for Left atrial enlargment diagnosis, AP diameter in PLAX view was the most used method (and not "has been").

Author Response

Thank you so much for taking the time to review this manuscript! Indeed, we are very proud of this work and we will seriously consider doing the meta-analysis you suggest.

I have taken all your recommendations into account. Please find the corresponding corrections highlighted in yellow in the attached file. I hope you find it interesting.

Reviewer 2 Report

Comments and Suggestions for Authors

Great Abstract. - No Comments

Introduction: I recommend providing more details on the atrial comorbidities in the introduction. Including types of AF (as most common). Consider including further literature on epidemiology to back up the introductory claims of why the Atria is important to understand. 

Atrial Anatomy & Physiology: At this level, anatomy and physiology can be further explained in advanced terminology. Consider including images. - Please include further discussion of Atria. I.e., LAA, irs occlusion, thrombus formation and clinical implications etc. 

Do not need to separate anatomy and physiology - this can be one paragraph. Consider including more details here. Consider adding a section to discuss the possible conditions and treatment options and their complexity. (refer to latest AHA or EU Heart Failure Guidelines)

Section 3: Please give more examples of advanced IAB, and its complications from the extensive registries or large databases if possible. 

No further comments. The discussion section seems acceptable in my opinion. 

Thank you for having me review this review article. Great insights! 

Author Response

Thank you so much for taking the time to review this manuscript. Please find the detailed responses below and the corresponding revisions highlighted in yellow in the re-submitted file.

a) Introduction: As you suggested, we have expanded the introduction by giving clinical relevance to the atrium by providing epidemiologic data.

b) Atrial Anatomy & Physiology: Following your suggestions, we have combined atrial anatomy and physiology. We have expanded the information by giving clinical relevance to the left atrial appendage with some therapeutic implications. We have also slightly expanded the information on normal atrial conduction.

c) Section 3: Please give more examples of advanced IAB, and its complications from the extensive registries or large databases if possible. 

Answer: In section 3, we focused on the definition, cause, classification, and diagnostic approach of the IAB. In section 4, we discussed Bayes' Syndrome and explained all the clinical consequences of IAB. However, as you suggested, in this section we have added information on the clinical relevance of IAB with data from major international registries.

Once again, thank you very much for your suggestions! They have clearly been important to improve the quality of our work.

Reviewer 3 Report

Comments and Suggestions for Authors

The authors made a pretty comprehensive review about interatrial conduction block, Bayes syndrome, left atrial enlargement and atrial failure. I generally agree with this review article. However, I think the concept about atrial failure still hazziness. IAB focuses on electrical properties and left atrial enlargement focuses on anatomical alternations. Both conditions may contribute to atrial fibrillation and even heart failure. I think knowing IAB, Bayes syndrome and LAE is enough to deal with a patient properly in clinical condition. Do we really have to define a term "atrial failure" ? 

My commet is to make the concept of atrial failure more clear and empowered this situation more clinical value. Otherwise, the authors can consider to just remove the paragraphs about atrial failure. 

Author Response

Thank you very much for taking the time to review this manuscript. Please find the detailed response below and the corresponding revision highlighted in yellow in the re-submitted file and at the bottom of this response.

Answer: Thank you so much for your comments on our paper and we clearly understand your opinion about this topic. We are aware that this is a new concept and that it may have certain flaws, however, we strongly believe that this new definition can promote research on atriopathy and its consequences, just as the authors of the great article published in the JACC (experts on this field) exposed in their paper.

We have added a paragraph to our paper referring to the relevance of this new definition in the form of "conclusions".

Once again we thank you for your comments and really hope that the changes we made in our paper seems interesting to you. 

6.3. Conclusions

Atrial failure should be considered in patients with compatible symptoms in the presence of atriopathy not attributable to ventricular or valvular pathology or extracardiac conditions. Advances in cardiac imaging and electrophysiology will allow us to improve our understanding of left atrial pathophysiology and its impact on global cardiac function. This new definition of atrial failure is likely to increase the volume of research focused on the left atrium in the future, with the goal of identifying therapeutic targets to prevent the clinical consequences of this entity.
